# Olfaction Recovery following Dupilumab Is Independent of Nasal Polyp Reduction in CRSwNP

**DOI:** 10.3390/jpm12081215

**Published:** 2022-07-26

**Authors:** Elena Cantone, Eugenio De Corso, Filippo Ricciardiello, Claudio Di Nola, Giusi Grimaldi, Viviana Allocca, Gaetano Motta

**Affiliations:** 1Department of Neuroscience, Reproductive and Odontostomatological Sciences—ENT Section, University of Naples Federico II, 80131 Naples, Italy; claudio.dinola@yahoo.it (C.D.N.); giusi.grimaldi90@gmail.com (G.G.); 2Head and Neck Department—ENT Section, AOU Federico II, 80131 Naples, Italy; 3Fondazione Policlinico Universitario A. Gemelli IRCCS, Head and Neck Surgery—Otorhinolaryngology, 00168 Rome, Italy; eugenio.decorso@gmail.com; 4ENT Department, AORN Cardarelli, 80131 Naples, Italy; filipporicciardiello@virgilio.it (F.R.); fess.federico@gmail.com (V.A.); 5Otorhinolaryngology, Head and Neck Surgery Unit, Department of Mental and Physical Health and Preventive Medicine, Università degli Studi della Campania Luigi Vanvitelli, 80131 Naples, Italy; gaetano.motta@unicampania.it

**Keywords:** CRSwNP, type 2, polyps, Dupilumab, olfaction, smell, biologics, QoL

## Abstract

Chronic rhinosinusitis with nasal polyps (CRSwNP) is a chronic type 2 inflammatory disease characterized by olfactory impairment (OI) as one of the most troublesome symptoms. Currently, biologics represent a new option in the treatment of uncontrolled type 2 CRSwNP. This is a retrospective real-life observational study involving adult patients affected by severe uncontrolled CRSwNP. At baseline, and 3 and 6 months after Dupilumab add on to intranasal steroids (INS), patients underwent the 22-item Sinonasal Outcome Test (SNOT-22), nasal endoscopy, Visual Analogue Scale (VAS) scale for OI, and Sniffin Sticks-16 items identification test (SS-I). We observed improvement in all clinical outcomes with a significant correlation between VAS-SS-I/SNOT22, whereas we did not find a correlation between Nasal Polyp Score (NPS) and SS-I or VAS. Interestingly, patients reported a higher degree of improvement of OI on the VAS than on the SS-I. These data demonstrate that the patients were not aware about the degree of their OI and the perception of general improvement in their health-related quality of life (HRQoL) may have influenced the VAS score. Moreover, we observed a lack of correlation between NPS and SS-I or VAS, suggesting that OI did not depend on the polyps’ volume and may be due mainly to the resolution of inflammation. So, the physiopathological mechanisms underlying OI in CRSwNP and its recovery after Dupilumab might be unrelated to the volume of the polyps and might depend mainly on the anti-inflammatory effects. Future studies including biomarkers may be useful to clarify this aspect.

## 1. Introduction

Chronic rhinosinusitis with nasal polyps (CRSwNP) is a chronic inflammatory disease characterized by long-term symptoms, such as nasal congestion, rhinorrhea/postnasal drip, olfactory impairment (OI), facial pain/pressure and presence of nasal polyps on nasal endoscopy and/or sinus opacification on computed tomography (CT) scan [1]. CRSwNP exhibits a prevalent type 2 inflammation in Western countries, characterized by interleukin (IL)-4, IL-5, and IL-13, and infiltration of nasal polyps by eosinophils, basophils, and mast cells [2,3].

CRSwNP is associated with significant impact on health-related quality of life (HRQoL). HRQoL is a multidimensional dynamic concept that includes physical, mental, and social domains that are influenced by disease and treatment [3]. According to the literature data, CRSwNP impacts multiple aspects of HRQoL and it is further worsened in patients with comorbidities, including asthma, non-steroidal anti-inflammatory drug-exacerbated respiratory disease (NSAID-ERD), or a history of sino-nasal surgery [3]. OI is one of the most troublesome and recalcitrant symptoms in patients with CRSwNP correlated with disease severity and HRQoL. It may be the first sign of disease recurrence [2].

OI is a multifactorial event in CRSwNP, involving different effects of chronic inflammation on the olfactory mucosa, edema of the neuroepithelium that inhibits the transmission of synaptic impulses (sensorineural hypothesis), and changes in airflow within the olfactory cleft (conductive hypothesis). To date, the mechanism of reversing OI in patients with CRSwNP is not well understood [4]. 

The standard of care for CRSwNP, intranasal steroids (INS), and repeated bursts/short courses of systemic/oral steroids (OCS) do not provide a long-lasting recovery of olfaction in subjects with CRSwNP [5] In addition, little evidence about olfactory outcomes between medical and surgical management of CRSwNP exists [2]. For instance, patients who benefit from surgery presented improvement in olfaction by 1 month that decreased by 3 months after surgery [6].

An ideal treatment for CRSwNP should achieve and maintain disease control, defined as absence of symptoms, improved HRQoL and health status, and improved endoscopic and radiologic outcomes. 

To date, biologics provided encouraging results in the clinical trials representing a possible new option in the treatment of uncontrolled CRS disease. 

Dupilumab is a fully human VelocImmune^®^-derived monoclonal antibody (Sanofi, Paris, France) that blocks interleukin (IL)-4Rα, the shared receptor component for IL-4 and IL-13, which are the key cytokines involved in type 2 mediated inflammation [7,8]. In the phase 3 SINUS-24 (NCT02912468) and SINUS-52 (NCT02898454) studies, Dupilumab added to the standard of care significantly reduced polyp size, sinus opacification, and severity of symptoms—among which OI—versus placebo and was generally well tolerated [9]. Dupilumab subcutaneous injection is the only biologic approved for the treatment of adults with severe uncontrolled CRSwNP in Italy [10] and the preliminary data in a real-life setting confirmed its efficacy improving quality of life and restoring olfactory associated disfunction [11].

Clinical trials and real-life data have already demonstrated the effects of Dupilumab vs. placebo on the recovery of olfaction, whereas what is still not clear is the mechanism of OI in CRSwNP and the physiopathological characteristics underlying the olfactory recovery after treatment [2,3,7,10,11].

The aim of this study is to comprehensively evaluate the impact of Dupilumab on olfactory outcomes in patients with severe CRSwNP and, at the same time, to investigate whether the mechanisms underlying OI in CRSwNP patients are related to nasal polyp volume.

## 2. Materials and Methods

This is a retrospective real-life observational study involving patients affected by severe uncontrolled CRSwNP. Subjects were selected from the medical records of patients in biological treatment with Dupilumab during follow-up at ENT Department of University Federico II and Cardarelli Hospital of Naples. Asthma was diagnosed according to GINA guidelines [12], and CRSwNP was diagnosed according to European Position Paper on Rhinosinusitis and Nasal Polyps (EPOS 2020) criteria [1]. 

Exclusion criteria were history of genetic, congenital, or acquired immunodeficiency, autoimmune diseases, current malignancy, previous diagnosis of OI, previous treatment with a different biologic, and previous radiotherapy for head and neck cancer. The study was conducted in accordance with Good Clinical Practice and with the principles ordained in the Declaration of Helsinki; the protocol was approved by local ethical board. All patients provided written informed consent. Patients 18 years of age or older with bilateral massive nasal polyps—NPS 5 out of maximum 8—and symptoms of CRS despite INS therapy were eligible if they had received 2 or more cycles of OCS per year, or long-term low-dose steroids in the previous year, or had a medical contraindication or intolerance to OCS, and at least one previous endoscopic sinus surgery (ESS). Type 2 inflammation was assessed at baseline with laboratory test for biomarkers in blood: eosinophils (EOS) and immunoglobulin (IgE) according to the EPOS 2020 cut-off [1]. After the enrollment, patients received 100 mg of mometasone furoate nasal spray in each nostril once daily and subcutaneous (SC) Dupilumab 300 mg every 2 weeks for 6 months. Enrolled patients were evaluated at baseline and at months 3 and 6. Polyp size were assessed by nasal endoscopy with a 2.7 mm 30 degree rigid endoscope (Storz, Tuttlingen, Germany) using the nasal polyp score (NPS). This score is graded based on nasal polyp size (recorded as the sum of the right and left nostril scores with a range of 0–8; higher scores indicate worse status) [13].

The sense of smell was assessed using a patient-reported visual analogue scale (VAS) for OI on a scale of 0 to 10 (higher scores indicate worse sense of smell) [1] and Sniffin Sticks-16 items identification test (SS-I) (Burghart, Wedel, Germany). The SS-I is a standardized and reliable test that consists of 16 common odors presented together with four verbal descriptors in a multiple forced-choice format (three distractors and one target). We considered a score ≥ 12 correct answers as normal [14,15]. To assess the HRQoL, 22-item Sinonasal Outcome Test (SNOT-22) was also performed. The SNOT-22 is a disease-specific HRQoL questionnaire evaluating the impact of CRS on HRQoL with a recall period of 2 weeks [16]. It includes 22 items, scored on a Likert-like scale categorized into 5 domains for CRSwNP: nasal, ear/facial, sleep, function, and emotion. The total score ranges from 0 to 110, higher scores represent worse HRQoL. The SNOT-22 domain scores allow for both an understanding of the burden of CRSwNP on a patient’s HRQoL, as well as a comprehensive assessment of a treatment’s effectiveness as well as objective measures of disease [17,18].

### Statistical Analysis

Continuous baseline characteristics were presented as mean and standard deviation. Correlation between parameters was measured using Pearson or Spearman rank order where appropriate. We tested the difference between the means by Student’s *t*-test and considered *p* < 0.05 to be statistically significant. Data were analyzed with SPSS v 27 for Windows (IBM Corp., Armonk, NY, USA). 

## 3. Results

In this retrospective observational study, we enrolled a cohort of 53 patients (20 females and 33 males; mean age: 53.07 ± 12.74) affected by severe uncontrolled CRSwNP and in ongoing treatment with Dupilumab in a real-life setting. Patient characteristics at the time of initiation of Dupilumab are shown in Table 1. Thirty-nine (74%) patients reported a history of allergy based on previous skin tests, 37 (70%) asthma, and no patient atopic dermatitis (Table 1).

Dupilumab was administered as add-on therapy to mometasone furoato nasal spray, 100 mg in each nostril once a day. No subject needed OCS therapy during biological treatment, except for one patient. 

All patients concluded the observation period of at least 24 weeks. Dupilumab was well tolerated. In only one case, we observed a hypereosinophilia (>3 × 10^9^/L) after 3 months of treatment in the absence of organ involvement, which returned < 1.5 × 10^9^/L after one cycle of oral prednisone 25 mg/day for 7 days, and subsequent tapering [17]

No patient reported conjunctivitis.

We observed that the mean SNOT-22 total score decreased from 55.6 ± 15 to 16.2 ± 12.5 after 3 months of therapy with a significant statistically difference (*p* = 0.0001). This improvement in HRQoL did not change even after 6 months, 15.9 ± 9.8 (*p* > 0.05) (Figure 1).

We also observed a statistically significant reduction in mean VAS score for OI from baseline 7.8 ± 2.5 to 2.5 ± 2.2 (*p* = 0.0001) at 3 months and 2.3 ± 2.1 at 6 months (*p* > 0.05) (Figure 1).

The NPS decreased from 5.8 ± 0.9 to 3.2 ± 1.2 (*p* = 0.0001) at 3 months and 1.6 ± 1.1 at 6 months (*p* = 0.0001) (Figure 1).

The SS-I demonstrated a statistically significant (*p* = 0.02) improvement of OI from 8.3 ± 3.6 at baseline to 11.5 ± 3.1 at 3 months and 11.9 ± 3.7 at 6 months (*p* > 0.05) (Figure 1).

In our series, the impact of Dupilumab on olfactory function was not influenced by age, gender, CRSwNP duration, polyps’ size, prior surgery, OCS use in the previous year, history of tobacco abuse, comorbid allergy and asthma or NSAID-ERD, and no statistically significant data were found among groups nor significant correlations (*p* > 0.05). 

We also observed a significant correlation between VAS score and SS-I at baseline (*p* = 0.03), 3 months (*p* = 0.03) and 6 months (*p* = 0.003).

We also found a significant correlation between SNOT22 and VAS at baseline (*p* = 0.01), 3 months (*p* = 0.001) and 6 months (*p* = 0.0001).

We did not find any correlation between the overall SNOT22 and SS-I, and between NPS and SS-I or VAS neither at baseline nor after therapy (*p* < 0.05).

Although the overall SNOT22 did not correlate with SS-I, we found a correlation between two items of SNOT22: decreased sense of smell/taste and frustration/ restlessness/irritability, and SS-I at baseline (*p* = 0.03; 0.04, respectively), 3 months (*p* = 0.03, 0.04, respectively) and 6 months (*p* = 0.02; 0.03, respectively).

## 4. Discussion

OI is one of the most troublesome and difficult-to-treat symptoms in CRSwNP. The evaluation of olfactory function is of utmost importance in the assessment of CRSwNP and it has been included in criteria for both patient selection and response to biologics in the European Forum for Allergy and Airway Diseases (EUFOREA) and the EPOS [1,2]. 

According to recent research, difficulty in recovering OI with standard therapy is an indicator of severe disease. For this reason, particular attention should be paid to this aspect; moreover, ENT specialists should be familiar with the most common diagnostic tools for the evaluation of OI in the clinical practice [19].

While the standard of care for CRSwNP seems to fail in making significant improvements in disease control, previous trials have demonstrated that Dupilumab can significantly and rapidly restore general CRSwNP symptoms and OI, with effects observed already within the first week and at the first assessment for olfactory function [2], 

According to the literature data, in our cohort we found a reversible OI in patients with CRSwNP following Dupilumab treatment, regardless of age, gender, CRSwNP duration, baseline polyp size, prior surgery, OCS use in the previous 2 years, history of tobacco abuse, allergy, comorbid asthma, or NSAID-ERD [2]. 

At baseline, we observed a significant impact of CRSwNP on HRQoL, as demonstrated by SNOT-22 and VAS scores; indeed, we found a VAS score for OI > 5 (7.8 ± 2.5), indicating a significant impact on the patients’ QoL [1]. 

The effect of Dupilumab on SNOT-22, VAS, and SS-I did not change from 3 months to 6 months. However, the NPS continued to significantly improve even at 6 months follow-up (Figure 1).

Very interestingly, although we observed a correlation between VAS and SS-I, the degree of improvement of olfactory function was higher by using the VAS score than using the SS-I, both at 3 months and 6 months (Figure 2).

These findings demonstrate that the patients were fully aware of their OI, but unexpectedly, patients were not aware about the degree of their OI. Most likely, patients tended to overestimate the improvement of OI. Probably because Dupilumab has a positive impact on the general HRQoL, improving all the aspects characterizing the CRSwNP, which positively influence patients’ lives and their ability to self-assessment the HRQoL. Briefly, patients may be influenced by a general improvement in their HRQoL and tended to emphasize all the outcomes obtained with treatment. This observation once again emphasizes the important role played by olfaction in terms of HRQoL in patients suffering from CRSwNP. These data might also suggest that even a minimal improvement in olfactory performance can improve the HRQoL of these patients. 

Thus, the improvement of VAS and SS-I and their correlation suggested that the VAS could be a reliable tool for self-assessment of OI, also for the purpose of saving time in outpatient assessment. However, the statistically significant difference observed between the degree of improvement in OI on the VAS score vs. that on the SS-I suggested that a semi-objective assessment performed by a specific test may be clinically useful in evaluating the efficacy of the biologic in a clinical real-life setting.

A very important clinical implication is that the VAS score alone seems to be not enough for the evaluation of OI, so SS-I is always necessary.

Although the overall SNOT-22 did not correlate with SS-I, we found a correlation between two items of SNOT-22, “decreased sense of smell/taste” and “frustration/restlessness/irritability”, and SS-I. Likely, the SNOT-22 questionnaire is disease-specific questionnaire including subjective and psychosocial domains that evaluate different characteristics of CRSwNP related QoL. Among these, the olfactory function represents only a part, albeit a very important one of the burdens of CRSwNP on the QoL. 

The correlation between item “frustration/restlessness/irritability” and SS-I, once again, underlines the strong impact of the olfactory function on QoL.

In addition, we did not find any correlation between NPS and SS-I or VAS, neither at baseline nor after therapy. These data suggest that OI in CRSwNP patients did not seem to depend directly on the size of the polyps, so that a volumetric reduction in polyps’ volume did not correlate with an improvement in the olfactory performance. It is therefore conceivable that the physiopathological mechanisms underlying OI in CRSwNP patients and the mechanisms underlying the olfactory recovery after therapy with Dupilumab are in part or completely unrelated to the volume effect of the polyps and might be mainly correlated to the resolution of local inflammation.

Furthermore, the successful treatment with Dupilumab in our sample size might imply that type 2 inflammation plays a role in the disease mechanism of OI in CRSwNP. 

In recent research by Mullol, the authors suggest that the degree of ongoing type 2 inflammatory processes affects the severity of olfactory dysfunction in CRSwNP [2].

However, the role of type 2 inflammation-mediated barrier dysfunction in the olfactory neuroepithelium remains to be determined [20]. Our speculations are in line with recent literature. It is conceivable that, at the basis of the OI in CRSwNP and of the olfactory recovery following treatment with Dupilumab, there are immunological mechanisms with the direct or indirect involvement of interleukins (ILs) and in particular IL-4, IL-5, IL-13, and eosinophils [20].

The olfactory neuroepithelium contains three major cell types of the peripheral sensory system, including olfactory sensory neurons (OSNs), that are particularly susceptible to local immune mediators in the setting of CRSwNP. Infiltrating immune cells in the olfactory neuroepithelium might account for the dysfunction of the peripheral olfactory system. So, the inflammation of the epithelium could affect olfactory neurogenesis, differentiation, and maturation of OSNs [20].

Previous studies found a reduction in mature OSNs, erosion of the olfactory neuroepithelium in CRS biopsy specimens, and high density of eosinophil infiltrations [21,22]. 

Eosinophilic infiltration may play a significant role in the OI in CRSwNP. In addition, increased levels of several cytokines in the olfactory cleft have been correlated with OI, elevated levels of IL-2, IL-5, IL-6, IL-10, and IL-13 are indeed associated with reduced test scores for smell identification and olfaction is found to be strongly correlated with levels of cytokines IL-5 and IL-13, although the role of the IL-4 and IL-13 pathways and their possible regulatory impacts on neurogenesis and homeostasis of olfactory neurons have not been determined yet [20].

Furthermore, the inflammatory natures of CRSwNP could be decisive also in the olfactory outcomes after the surgery. 

Although surgery may be beneficial OI, the results of olfactory function after surgery are inconsistent in CRSwNP patients and the improvement of OI with time among eosinophilic CRSwNP patients seems not to be well-sustained after surgery [23]. 

However, the administration of adjuvant medical therapy post-operatively may aid the continued recovery of olfactory function [20].

These observations agree with other studies in the literature in which elevated levels of in olfactory cleft mucus are associated with reduced olfactory identification scores in CRS patients and altered levels of select olfactory mucus ILs are potentially deleterious for the olfactory neuron function and turnover [4]. 

In conclusion, our preliminary data confirmed the efficacy of Dupilumab in a real-life study. Our data demonstrated that the patients may be influenced in subjective estimation of olfactory recovery by the perception of general improvement in their HRQoL. Moreover, we observed that OI did not depend on the polyps’ volume and might be due mainly to the resolution of inflammation. The current study findings should be interpreted considering some methodological limitations especially about the monocentric design of the study, small sample size, and 6 months follow-up. Our findings encourage future studies to investigate whether Dupilumab’s recovery of olfaction in CRSwNP patients may be primarily dependent on anti-inflammatory effects. Future studies that include biomarkers could be useful to clarify this aspect.

## Figures and Tables

**Figure 1 jpm-12-01215-f001:**
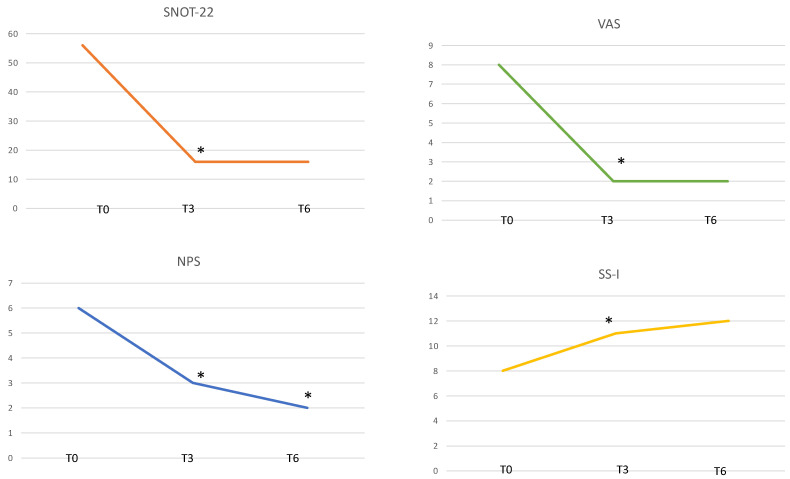
Clinical outcomes. The mean 22-item Sinonasal Outcome Test (SNOT-22 )total score, visual analogue scale (VAS), nasal polyp score (NPS), and Sniffin Sticks-16 items identification test (SS-I) at T0, T3 and T6. * Statistical significance. T0: baseline; T3: 3 months follow-up; T6: 6 months follow-up.

**Figure 2 jpm-12-01215-f002:**
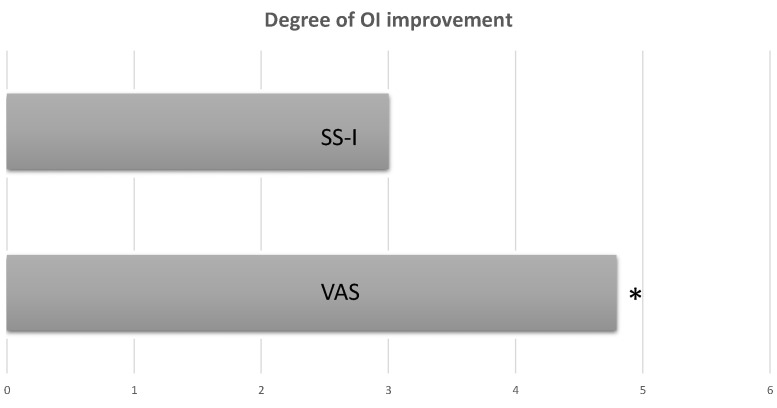
Degree of improvement of olfactory impairment (OI). Higher degree of improvement of OI on the VAS score than on the SS-I. * Statistical significance. T0: baseline; T3: 3 months follow-up; T6: 6 months follow-up.

**Table 1 jpm-12-01215-t001:** Patient characteristics at baseline.

*N.ro*	53
*Sex*	33 (62.2%) M; 20 (37.8%) F
*Age*	53.07 ± 12.74
*Smokers*	8 (15%)
*NSAID-ERD*	11 (21%)
*Allergy*	39 (74%)
*Asthma*	37 (70%)
*N.ro surgeries*	15 (28%) > 2 surgeries

## Data Availability

Data are available upon reasonable request.

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
