# Peer review of "Olfaction Recovery following Dupilumab Is Independent of Nasal Polyp Reduction in CRSwNP"

_jpm, 2022, doi:10.3390/jpm12081215_

Round 1

Reviewer 1 Report

The authors provide a detailed analysis of variables monitored as part of several IRB trials assessing he impact of an immunobiological, dupilumab, on chronic rhinosinusitis with nasal polyposis.

The authors should provide the "allergic sensitization status" of the patients (skin testing or serology) to provide another variable for the potential impact of the treatment.  

In addition, should provide provide the co-morbid status that would include asthma and atopic dermatitis.

Were there any adverse effects noted e.g. conjunctivitis?

Were any specific fields in the SNOT-22 that were more statistically noted ? A breakdown of the individual question and changes would be very helpful versus just using the overall SNOT-22 score. The SNOT-22 includes both subjective and psychosocial domains. Which specific fields had more of an impact over time of treatment? The lack of correlation of SNOT 22 and שבת שלום_I is expected as SNOT as mutipe other non smell based  questions (21 others). Question 5 in the SNOT would be a direct link to שבת שלום-! as they both relate to smell.

Unclear as to why  the title reflects "unexpected" when the trial based on preliminary data was expected to have an effect on CRSwNP?

In addition the authors do not provide any measurement of inflammatory mediators to justify the finding of "anti-inflammatory effect".  

Define all abbreviations when used for the first time. INS, SNOT, VAS, NPS.

Author Response

Reviewer 1

Thank you for your precious advice for the revision.

  1. The authors should provide the "allergic sensitization status" of the patients (skin testing or serology) to provide another variable for the potential impact of the treatment.

In addition, should provide provide the co-morbid status that would include asthma and

atopic dermatitis.

Allergic status and comorbidities are in Tab.I. In addition, we added the allergic status and other comorbidities in the text:

“Thirthy-nine (74%) patients reported a history of allergy, based on previous skin tests, 37 (70%) asthma, and no patient atopoic dermatitis.

In our series the impact of Dupilumab on olfactory function was not influenced by age, gender, CRSwNP duration, polyps’ size, prior surgery, OCS use in the previous year, history of tobacco abuse, comorbid allergy and asthma or NSAID-ERD, no statistically significant data were found among groups, nor significant correlations.

  1. Were there any adverse effects noted e.g., conjunctivitis?

No patient reported conjunctivitis.

  1. Were any specific fields in the SNOT-22 that were more statistically noted? A breakdown

of the individual question and changes would be very helpful versus just using the overall

SNOT-22 score.

 We added these observations in

Results

Although the SS-I did not correlate with the overall SNOT22, we found a correlation between two items of SNOT22, decreased sense of smell/taste and frustration / restlessness / irritability, and SS-I at baseline (p=0.03; 0.04 respectively), 3 months (p=0.03, 0.04 respectively), and 6 months (p=0.02; 0.03 respectively).

 Discussion

Although the overall SNOT22 did not correlate with SS-I, we found a correlation between two items of SNOT22, “decreased sense of smell/taste” and “frustration/restlessness/irritability”, and SS-I. Likely, the SNOT-22 questionnaire is disease-specific questionnaire including subjective and pshycosocial domains that evaluate different characteristics of CRSwNP related QoL. Among these, the olfactory function represents only a part, albeit a very important one of the burdens of CRSwNP on the QoL.

 The correlation between item “frustration/restlessness/irritability” and SS-I, once again underlines the strong impact of the olfactory function on QoL.

  1. Unclear as to why the title reflects "unexpected" when the trial based on preliminary data

was expected to have an effect on CRSwNP?

The term "unexpected" refers to the lack of correlation between hyposmia and polyp volume.

However, we changed the title as follows:

Olfaction recovery following Dupilumab is independent of nasal polyps’reduction in CRSwNP

  1. In addition, the authors do not provide any measurement of inflammatory mediators to

justify the finding of "anti-inflammatory effect".

This is a real-life study, and the anti-inflammatory effect of Dupilumab on OI is speculative and derives from literature data. However, the anti-inflammatory effect of biologics on olfactory recovery should be studied, first, "in vitro", as it involves many mediators, some of which have not yet been identified.

Discussion

             In addition, this is a real-life study and the physiopathology of OI in CRSwNP and the anti-inflammatory effect of Dupilumab on OI are still speculative. Thus, the anti-inflammatory effect of biologics on the olfactory recovery should be better studied, first, "in vitro", as it involves many mediators, some of which have not yet been identified.

……In conclusion, our preliminary data confirmed the efficacy of Dupilumab in a real-life study. Our data demonstrated that the patients may be influenced in subjective estimation of olfactoy recovery by the perception of general improvement in their HRQoL. Moreover, we observed that OI did not depend on the polyps’ volume and might be due mainly to the resolution of inflammation. The current study findings should be interpreted considering some methodological limitations especially about the monocentric design of the study, small sample size, and 6-months follow up. Our findings encourage future studies to investigate whether Dupilumab's recovery of olfaction in CRSwNP patients may be primarily dependent on anti-inflammatory effects. Future studies that include biomarkers could be useful to clarify this aspect.

 Hoping to have implemented your advice correctly and confiding to have improved substantially the scientific message of the paper.

my best regards,

the corresponding author

Elana Cantone

Reviewer 2 Report

In this study the main finding is the difference between two subjective measurements such as VAS and SS-I at baseline, at 3 months and at 6 months which was not affected from any other characteristic of the patiients.  All other findings were related to good response due to dupilumab administration. In the article there is no actual explanation for this difference between VAS and SS-I among patients.

There are no comments whether or not this difference is clinically significant and if the SS-I or the VAS is more important to use it as a tool, in clinical practice based on the  findings from this study. Also there is no correlation mentioned between the findings and the underlying mechanisms, although it was one of the aims of the study.

Author Response

Reviewer 2

Thank you for your precious advice

In this study the main finding is the difference between two subjective measurements such as VAS and SS-I at baseline, at 3 months and at 6 months which was not affected from any other characteristic of the patients.  All other findings were related to good response due to dupilumab administration. In the article there is no actual explanation for this difference between VAS and SS-I among patients. There are no comments whether this difference is clinically significant and if the SS-I or the VAS is more important to use it as a tool, in clinical practice based on the findings from this study. 

We found that both VAS and SS-I improved after treatment. In addition, we found a statistically significant correlation between VAS score and SS-I at baseline, 3 months, and 6 months, the only difference observed was a higher degree of improvement in olfactory function on the VAS score than on the SS-I. Patients might be influenced by a general improvement in their HRQoL and tended to emphasize all the outcomes obtained with treatment…These data might also suggest that even a minimal improvement in olfactory performance can improve the HRQoL of these patients.

The most important clinical implication is that a simple VAS is not enough for the evaluation of OI, but SS-I is always necessary. We stressed this concept in discussions.

Also, there is no correlation mentioned between the findings and the underlying mechanisms, although it was one of the aims of the study.

Thank you for your suggestion, now we have better clarified the aim of the study as follows:

“The aim of this study was to comprehensively evaluate the impact of Dupilumab on olfactory outcomes in patients with severe CRSwNP and, at the same time, to investigate whether the mechanisms underlying OI in CRSwNP patients are related to nasal polys’ volume.”

 Hoping to have implemented your advice correctly and confiding to have improved substantially the scientific message of the paper.

my best regards,

the corresponding author

Elana Cantone